# Magnesium Depletion Score as an Indicator of Health Risk and Nutritional Status—A Scoping Review

**DOI:** 10.3390/nu17203286

**Published:** 2025-10-20

**Authors:** Rebecca B. Costello, Zhongqi Fan, Taylor C. Wallace

**Affiliations:** 1Center for Magnesium Education and Research, Pahoa, HI 96778, USA; 2Think Healthy Group, LLC, Washington, DC 20001, USA; elienafan@hotmail.com; 3School of Medicine and Health Sciences, George Washington University, Washington, DC 20052, USA; 4Gerald J. and Dorothy R. Friedman School of Nutrition Science and Policy, Tufts University, Boston, MA 02155, USA

**Keywords:** magnesium, magnesium depletion score, magnesium deficiency, nutritional status

## Abstract

**Background/Objectives**: Magnesium is an essential nutrient involved in more than 600 enzymatic reactions, and nutritional status is estimated to be critical for many metabolic and biochemical processes in humans. Although magnesium deficiency and inadequacy impacts multiple chronic disease states, signs and symptoms are often nonspecific and nutritional status is difficult to measure. The recently developed magnesium depletion score (MDS) is a promising tool for identifying individuals at risk of magnesium deficiency or inadequacy and associated comorbidities, but its clinical applicability and validity across broad populations remains unclear. **Methods:** Using the Joanna Briggs Institute (JBI) and JBI Collaborating Center guidelines for conducting scoping reviews, four electronic databases (MEDLINE/ PubMed, Embase, and Scopus) were systematically searched from inception to 20 May 2025 for clinical and observational English-language studies that assessed the impact of MDS on health and/or nutritional status. The protocol was preregistered on Open Science Framework prior to data extraction. **Results:** 48 articles, inclusive of 39 cross-sectional and 15 prospective cohort analyses, as well as a single secondary analysis of a randomized controlled trial, were included in the scoping review. All but two analyses reported adverse associations with a high MDS. MDS was inversely correlated with dietary magnesium intake across studies. **Conclusions:** The MDS, particularly when utilized alongside traditional dietary intake assessment, offers promise as a tool for more rapidly identifying individuals at risk of magnesium deficiency (or insufficiency), and associated comorbidities, although large clinical trials are needed to confirm these findings.

## 1. Introduction

Magnesium is an essential nutrient that is involved in numerous metabolic and biochemical process within the cell; the mineral is thought to serve as an activator and cofactor for >200 and >600 enzymes in the human body, respectively [1,2,3]. A large portion of the population fails to consume current recommended intakes of magnesium from food alone (e.g., green vegetables, nuts, seeds, beans, and whole grains) and subsequently are at risk for suboptimal magnesium status; >50% of Americans today do not consume the estimated average requirement (EAR) for magnesium according to recent analysis of the U.S. National Health and Nutrition Examination Survey [4].

Strong evidence suggests that magnesium deficiency contributes to cardiovascular disease [5,6], the metabolic syndrome and type 2 diabetes [7,8], and osteoporosis [9]. Magnesium insufficiency may likely be a contributor to these long-term health ailments, as it has been consistently linked to increases in subclinical low-grade chronic inflammation. Along these lines, evidence from epidemiological analyses and/or clinical trials suggest regular nut and/or legume consumption (two sources high in magnesium) to have beneficial impacts on multiple chronic disease states, as well as in reducing oxidative stress, inflammation, visceral adiposity, hyperglycemia, insulin resistance, and endothelial dysfunction [10].

The signs and symptoms of magnesium deficiency and insufficiency are numerous, nonspecific, and widespread [11]. However, magnesium status is difficult to measure and may be best defined by dietary intake coupled with serum magnesium concentrations and urinary magnesium excretion [12].

The magnesium depletion score (MDS) is an aggregate of several risk factors affecting the absorption and excretion of magnesium. The MDS has been used to identify individuals with abnormal magnesium absorption and/or excretion that may result in a deficient magnesium status. The score was originally derived from the assessment of nutritional status using the magnesium load retention study [13]. This test determines the percent of a magnesium load that is retained by the body by measuring the percent of the load that is excreted in the urine within 24–48 h of administration. It is currently the only tool to assess adequacy of body magnesium stores. Following development of the MDS, the same author group then validated it against a cohort of participants enrolled in the U.S. National Health and Nutrition Examination Survey (NHANES) for greater applicability [13]. The MDS is noninvasive and is calculated as a composite of the following factors:Current use of diuretics counted as 1 point.Current use of proton pump inhibitor (PPI) counted as 1 point.Heavy drinker (defined as >1 drink/d for women and >2 drinks/d for men) counted as 1 point.Mildly decreased kidney function, defined as estimated glomerular filtration rate ≥ (eGFR) 60 mL/(min × 1.73 m^2^) < eGFR 90 mL/min × 1.73 m^2^, counted as 1 point.Chronic kidney disease defined as eGFR < 60 mL/min × 1.73 m^2^ counted as 2 points.

An MDS >2 has been used to indicate magnesium deficiency associated with increased risk for systemic inflammation and cardiovascular mortality in adults [13]. A score of >3 combined with a dietary magnesium intake below the US recommended dietary allowance (RDA) has been used to indicate magnesium deficiency associated with osteoporosis [9]. These data suggest that sample size and/or disease entity may influence the cut-point indicative of magnesium deficiency. This new suggested method of magnesium status assessment, especially for individuals with diseases and/or ailments associated with magnesium deficiency, needs further evaluation and validation before being accepted for general use.

This scoping review sought to answer the following question: What is the extent of available evidence investigating the application of the magnesium depletion score as an indicator of health risk and nutritional status?

## 2. Materials and Methods

We utilized the Joanna Briggs Institute (JBI) and JBI Collaborating Center guidelines for conducting scoping reviews [14,15] and report results per the Preferred Reporting Items for Systematic Reviews and Meta-Analyses (PRISMA) Extension for Scoping Reviews (PRISMA-ScR) checklist [16]. We also followed the suggested framework by Arksey and O’Malley [17], which consists of the following components: defining the review question, identifying relevant articles, charting the data, and summarizing the findings. The Population–Concept–Outcome (P-C-O) approach was used to assist in the structure of our search strategy and eligibility criteria. The protocol was preregistered on Open Science Framework (https://osf.io/vka6h) prior to data extraction.

### 2.1. Data Sources and Search Strategy

A trained librarian within the George Washington University Himmelfarb Health Sciences Library assisted a study investigator (T.C.W.) in developing and implementing a comprehensive web-based search of the MEDLINE/PubMed (National Library of Medicine, Bethesda, MD, USA), Embase (Wiley, West Sussex, UK), Web of Science (Clarivate, Philadelphia, PA, USA), and Scopus (Elsevier, Mérignac, France), databases from inception to 20 May 2025. The complete search strategy is provided in Appendix A.

### 2.2. Study Selection and Data Extraction

Search results for each database were downloaded and imported into Rayyan AI software (Rayyan Systems Inc.; Cambridge, MA, USA, https://www.rayyan.ai), where duplicates were detected primarily through article DOIs and removed prior to screening. Articles were screened to prespecified eligibility criteria presented in Appendix A. This scoping review included peer-reviewed and English-language clinical trials and observational studies that assessed potential relationships between the newly proposed MDS on human health outcomes. We did not restrict eligibility based on study duration, participant age, participant health status, health outcomes, or date of publication. Independent dual title and abstract screening was conducted (Z.F. and T.C.W.), with conflicts being resolved through consensus. Remaining articles underwent similar independent dual full-text screening (Z.F. and T.C.W.) utilizing the same inclusion and exclusion criteria. The investigators met to discuss and reconcile any discrepancies through consensus. Reference lists of all included articles were hand-searched prior to data extraction to ensure retainment of all relevant articles. Standardized data-extraction forms were created in Microsoft Excel (version 16.93.1; Microsoft, Redmond, WA, USA) to extract information on study design, population, sample size, intervention/exposure, estimated magnesium intake (in mg/d), duration or follow-up, main outcomes, and overall findings related to MDS. One investigator (Z.F.) extracted data from all included articles, after which a second investigator (T.C.W.) quality checked all extracted data to ensure their accuracy. Discrepancies were resolved through consensus between the two investigators. Descriptive statistics were calculated using Microsoft Excel software. Number (n) and percent frequency (%) are used to describe categorical variables.

## 3. Results

### 3.1. Characteristics of Included Articles

Our literature search strategy identified 66 articles for title and abstract screening, after the removal of duplicates. Of these articles, 48 met our eligibility criteria and moved forward to full-text screening. All 48 articles screened in full text met our eligibility criteria and were included in the scoping review. These articles contained 39 cross-sectional and 15 prospective cohort investigations, as well as a single secondary analysis of a randomized controlled trial (Table 1 and Table 2). Data cycles from NHANES represented within all 48 articles with results of the secondary analysis of a clinical trial also being co-published alongside the Fan et al., 2021 [13] NHANES analysis. Figure 1 provides the PRISMA flow diagram of studies.

The NHANES data cycles examined varied, with the widest spread being 1988–2018 for a prospective study by Xing et al. [59]. One cross-sectional study covered only one NHANES data cycle for years 2013–2014 in adults aged >60 years [31], and four articles included both cross-sectional and prospective data reported from 10 cycles (1999–2018) [26,47,48,53]. The largest sample size included 44,588 adults (aged ≥18 years) [48] over 10 NHANES cycles. The age ranges varied as well but only included adults aged ≥18 years, and several articles enrolled participants at age ≥40 years. One prospective analysis by Jiang et al. [26] enrolled older adults (aged >60 years) with frailty. The duration of prospective cohort analyses ranged from 12 weeks to a median of 31 years. The largest prospective analysis enrolled 16,394 adults from 1999 to 2018 NHANES data cycles [54].

### 3.2. MDS Scoring Parameters

The MDS scoring parameters (0–5) also varied across included articles. Twenty-seven articles reported individual MDSs of 0 to ≥3. Two articles reported individual MDSs of 0 to ≥4, and 10 articles reported individual MDSs of 0 to 5. Four articles used a broad scoring category of <2 or >2. As the individual MDSs increased from 0 to 5, the percentage of participants in each category decreased.

Because the GFR is a key component of the MDS, the majority of articles utilized the Chronic Kidney Disease Epidemiology Collaboration (CKDEPI) equation by Levy et al. [62] updated in 2021, as included in the NHANES. For scoring alcoholic beverage consumption, the majority of articles utilized the NHANES questionnaire derived from the 2015–2020 Dietary Guidelines for Americans Food Patterns Equivalent database [63]. The quantity of alcohol intake was defined as 1–2, 3–4, and ≥5 drinks/drinking day for men and 1, 2–3, and ≥4 drinks/drinking day for women. According to NHANES analytic guidelines, reports of <1 drink/drinking day were rounded up and coded as 1 drink/drinking day.

### 3.3. Health Outcomes

Health outcomes across observational analyses were multifaceted, ranging from mortality, disease, and disease event risk to self-reported sleep quality and biomarker measures (e.g., high-sensitivity C-reactive protein). The most studied health outcomes were all-cause and cardiovascular mortality and cardiovascular disease or related biomarker measures. Table 3 presents characteristics of analyses investigating the effects of MDS on health outcomes.

Only three cross-sectional analyses failed to find a beneficial relationship between lower MDS and health or nutritional status. Two of these cross-sectional analyses failed to show a relationship with MDS on serum klotho levels [34,52], and the third failed to show a relationship with depression [18]. Two prospective analyses showed a high MDS to be associated with cardiovascular and all-cause mortality but failed to show any relationship with cancer mortality [55,60].

Dietary magnesium intakes were reported in most articles that utilized data cycles from the U.S. NHANES (Table 4). Twenty-three articles reported magnesium intakes by MDS, and all articles providing mean dietary intake levels showed suboptimal magnesium intake below the EAR (<350 magnesium/d) at all MDS levels. Twenty-three articles provided a subgroup analysis of magnesium intakes based on disease or health condition at baseline (e.g., hypertension vs. non-hypertensive), and 18 articles evaluated intakes by cut-offs based on the EAR, RDA, tolerable upper intake level (UL), or median intakes. Only seven articles reported magnesium intake from dietary supplements, with all indicating suboptimal intake from total diet (food + dietary supplements) [13,22,36,45,55,56,58]. Thirteen articles commented on lower magnesium intake being associated with higher MDSs.

## 4. Discussion

This scoping review highlights the MDS as a new methodology for the determination of magnesium deficiency based on five criteria, and a higher MDS denotes a greater degree of magnesium deficiency. The MDS methodology was applied across 48 articles using well-established NHANES criteria and methodology, data collection, standardized questionnaires (ethyl alcohol intake), and GFR values determined by standardized protocol. Findings of this scoping review suggest that the MDS methodology can serve as a valid tool to assess magnesium deficiency, as it has demonstrated reproducibility and is highly correlated with disease outcomes. The MDS is noninvasive and cost-effective compared to biochemical tests and integrates multiple risk factors, offering a holistic assessment of nutritional status. However, the MDS may better reflect long-term magnesium status compared to serum levels alone, which can fluctuate acutely. There is also a need to appreciate the numerous factors affecting serum magnesium concentrations when considering the reliability of this as a measure of magnesium status, such as diurnal variation, strenuous exercise, various medications, and disease states [5]. It should be noted, however, that the MDS is heavily dependent on kidney function, with the GFR contributing 1 to 2 points; and GFR decreases with age. In addition to chronic kidney disease, high blood pressure, and diabetes, a decrease in GFR is also indicative of disorders such as microinflammation, endothelial dysfunction, oxidative stress, and increased aortic pressure [64]. Liu and colleagues [29] recently demonstrated that dietary magnesium intake and GFR were inversely correlated with risk of stroke, and participants with low dietary magnesium intake had higher stroke risk than those with normal (>254 mg/d) magnesium intake. In this scoping review, two cross-sectional analyses [23,48] evaluating stroke outcomes found that increasing MDS was associated with an increased risk of stroke in individuals with low dietary intakes of magnesium (<254 mg/d). In a prospective study, Xing [59] found that heavy drinking was the most influential factor among the four MDS scoring items that affected mortality outcomes in patients with kidney disease (GFR < 60) and these patients had the lowest mean survival time. Use of PPIs and diuretics was not as highly correlated with survival time. This finding suggests that sample size or disease entity may influence the MDS cut-point indicative of magnesium deficiency and needs further evaluation and validation before being accepted for general use [5].

Regarding the health outcomes under study, the data consistently demonstrated a higher MDS to be associated with an increased risk of all-cause and cause-specific mortality, as well as numerous biomarkers, surrogate endpoints, and chronic disease outcomes. We note the consistent relationship between a higher MDS with increased risks of outcomes known to be associated with low magnesium intake and/or status, such as CVD and hypertension, CKD and impaired kidney function, diabetes and glucose–insulin dynamics, and related biomarkers. Higher MDS was also associated with elevated hs-CRP levels, consistent with the existing scientific literature linking low magnesium intake or status to increased hs-CRP. The scoping review also identified correlations between a higher MDS and several outcomes not traditionally associated with low magnesium intake or status, such as gout and periodontitis. It is possible that a higher MDS reflects poor overall health rather than being causally related to the health outcomes described in this scoping review of observational data from NHANES (see Limitations section).

### Limitations

The findings reported here are affected by the limitations of the literature included in this review. Although the data were collected from large datasets, there is some heterogeneity in reporting the results: some authors chose to report means for all components of the MDS tool, and most did not. The MDS scoring methodology varied among included articles, with some reporting scales of 0–2 and others reporting on a scale of 0–5. Some authors chose to report MDS by disease and non-disease subgroups. This scoping review indicates a lack of prospective cohort analyses (aside those using the U.S. NHANES and U.S. National Death Index) of sufficient duration and a lack of RCTs testing whether MDS-guided interventions (e.g., magnesium supplementation, reduced PPI use) improve clinical outcomes. Thus, the findings are limited to NHANES cohorts with varying duration of follow-up, the dietary intake measures were stratified by varying means of classification, and the analyses were conducted exclusively in adults aged ≥18 years. Importantly, prospective analyses of NHANES are limited to all-cause and major disease mortalities recorded in sufficient numbers within the U.S. National Death Index (e.g., CVD, cancer). This constrains our ability to determine which disease outcomes are most influenced by a higher MDS, beyond the evident elevated risk of CVD mortality and the comparatively weaker or null association with cancer mortality. These types of NHANES analyses are further limited by reliance on a single baseline calculated MDS rather than repeated measures widely available in other prospective cohorts.

## 5. Conclusions

This scoping review synthesized existing evidence reporting the clinical applicability and validity of MDS across broad populations. There is consistent evidence from existing observational studies that demonstrate a high MDS is associated with adverse health status in humans. Higher MDS was also shown to be inversely correlated with dietary magnesium intake across existing observational studies. The MDS, particularly when utilized alongside traditional dietary intake assessment, offers promise as a tool for more rapidly identifying individuals at risk of magnesium deficiency (or insufficiency), and associated comorbidities, although large clinical trials are needed to confirm these findings.

## Figures and Tables

**Figure 1 nutrients-17-03286-f001:**
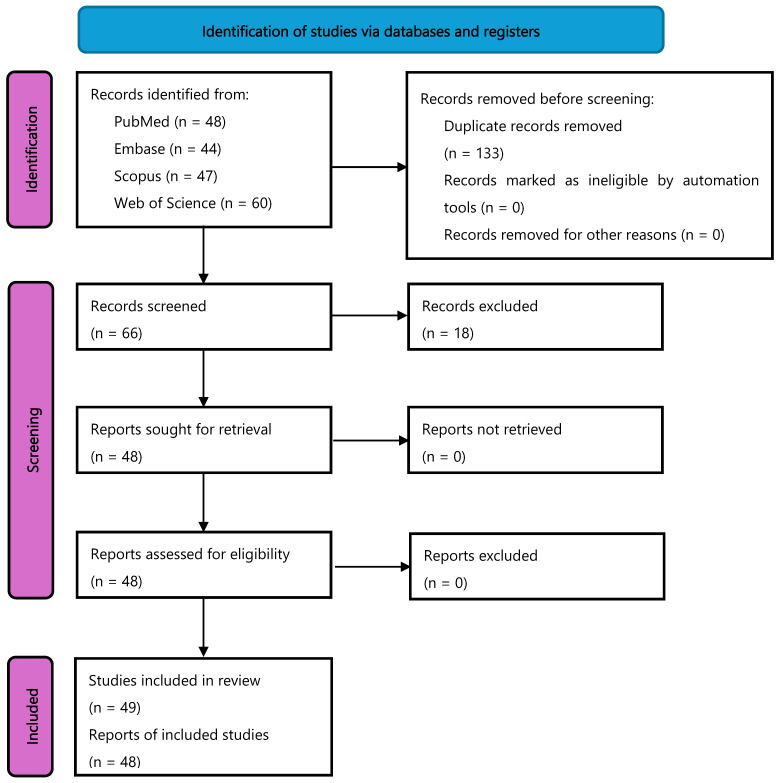
PRISMA flow diagram of included articles.

**Table 1 nutrients-17-03286-t001:** Characteristics of included cross-sectional analyses using the MDS.

Reference	Sample Size and Population	Age (y), Mean (SD) or Median (Range)	Intervention or Exposure (MDS)	Magnesium Intake (mg/d)	Main Outcomes	No. of Events	Main Findings
Cai et al. (2024) [18]	Adults (20–60 y) enrolled in the 2005–2018 NHANES data cycles (*n* = 18,247)	40.69	MDS 0	308.61 ± 2.03	Depression	1753	MDS may be positively associated with risk of depression (as diagnosed by PHQ-9)
			MDS 1				
			MDS 2				
			MDS ≥3				
Cai et al. (2025) [19]	Adults (≥40 y) enrolled in the 1999–2004 NHANES data cycles (*n* = 6571)	56.13 (0.23)	MDS 0	285.10 ± 3.06	PAD	NR	↑ MDS was potentially linked to an increased risk of PAD. Individuals aged >60 y and those with MDS ≥3 may be at heightened risk of PAD
			MDS 1				
			MDS 2				
			MDS ≥3				
Cao et al. (2024) [20]	Adults (age NR) enrolled in the 2007–2016 NHANES data cycles (*n* = 18,039)	40.96	MDS 0	273.0 (198.0–371.0)	Gout	851	↑ MDS associated with ↑ gout risk. Dietary magnesium intake did not moderate the correlation between MDS and gout risk
			MDS 1				
			MDS 2				
			MDS ≥3				
Cen et al. (2024) [21]	Adults (≥40 y) enrolled in the 2003–2018 NHANES data cycles (*n* = 20,010)	57.6	MDS 0	298.70 ± 2.82	Parkinson’s disease	240	↑ MDS associated with ↑ risk of Parkinson’s disease. Each increase in 1 unit in MDS was liked to ~50% higher probability of Parkinson’s disease. Individuals in the middle and high MDS groups are at a higher risk of Parkinson’s disease
			MDS 1				
			MDS 2				
			MDS ≥3				
Chen et al. (2023) [22]	Individuals (assumed adult; age NR) with diabetes mellitus enrolled in the 2005–2018 NHANES data cycles (*n* = 4308)		MDS 0–1	NR	Diabetic retinopathy	898	↑ MDS associated with ↑ risk of diabetic retinopathy
			MDS 2				
			MDS >2				
Fan et al. (2021) [13]	Adults (≥20 y) enrolled in the 2005–2010 NHANES data cycles (*n* = 11,693), 3 cycles		MDS 0	288 (median)	hs-CRP	NA	↑ MDS associated with ↑ risk of hs-CRP >3.0
			MDS 1				
			MDS 2				
			MDS >2				
Feng et al. (2024) [23]	Adults (≥18 y) diagnosed with CKD enrolled in the 2009–2016 NHANES data cycles (*n* = 3536)		Low MDS	275.53 ± 3.16	Stroke	359	↑ MDS associated with ↑ risk of stroke in CKD patients
			Medium MDS				
			High MDS				
Gong et al. (2025) [24]	Adult males (≥20 y) enrolled in the 2005–2018 NHANES data cycles (*n* = 16,043), 7 cycles		MDS 0	NR	Prostate cancer	1994	Significant association between prostate cancer risk, with ↑ MDS linked to ↑ prostate cancer prevalence
			MDS 1				
			MDS 2				
			MDS ≥3				
Gong et al. (2025) [24]	Male participants (≥20 y) enrolled in the 2005–2018 NHANES data cycles, 7 cycles (*n* = 16,043)		MDS 0	NR	Prostate cancer	511	Significant association between MDS and prostate cancer risk, with a higher MDS linked to increased prostate cancer prevalence
			MDS 1				
			MDS 2				
			MDS ≥3				
Gong et al. (2025) [25]	Adults (≥20 y) enrolled in the 2005–2018 NHANES data cycles (*n* = 32,493), 7 cycles		MDS 0	NR	Overactive bladder	6716	Significant positive association between MDS and overactive bladder
			MDS 1				
			MDS 2				
			MDS ≥3				
Jiang et al. (2025) [26]	Adults (>60 y) enrolled in the 1999–2018 NHANES data cycles (*n* = 13,3551)	71.31	MDS 0	264.03 ± 2.67	Prevalence of fragility	4464	↑ MDS is related to ↑prevalence of frailty in US older adults
			MDS 1				
			MDS 2				
			MDS ≥3				
Li et al. (2024) [27]	Adults (≥20 y) enrolled in the 2005–2018 NHANES data cycles (*n* = 12,023)		MDS	NR	MASLD	NR	↑ MDS associated with ↑ risk of MASLD
			MDS 1				
			MDS 2				
			MDS 3				
			MDS 4				
			MDS 5				
Li et al. (2025) [27]	Adults (≥20 y) enrolled in the 2003–2018 NHANES data cycles (*n* = 12,540)	50 (35–64)	MDS 0–1	267.00 (median)	Hyperuricemia	2466	↑ MDS associated with ↑ risk of hyperuricemia
			MDS 2				
			MDS 3–5				
Li et al. (2024) [28]	Adults (≥20 y) enrolled in the 2007–2018 NHANES data cycles (*n* = 20,513), 6 cycles		MDS 0	NR	RA	848	↑ MDS associated with ↑ odds of having RA and ↑ OA
			MDS 1		OA	2812	
			MDS 2				
			MDS 3				
			MDS ≥4				
Liu et al. (2024) [29]	Older adults (≥60 y) enrolled in the 2009–2018 NHANES data cycles (*n* = 3383), 5 cycles		MDS 0–1	NR	Anemia	382	↑ MDS associated with ↑ risk of anemia
			MDS 2				
			MDS ≥3				
Liu et al. (2025) [30]	Gout patients admit- ted to a rheumatology hospital in Sichuan, China, between February 2023 and February 2024 (*n* = 502)		MDS 0	NR	KSR	NR	MDS was significantly and positively correlated with the prevalence of KSR in gout patients. MDS appeared to mediate the association between serum uric acid and kidney stones
			MDS 1				
			MDS 2				
			MDS ≥3				
Lu et al. (2023) [31]	Adults (≥40 y) enrolled in the 2013–2014 NHANES data cycles (*n* = 2640)		MDS 0	288.76	AAC score	NA	↑ MDS associated with ↑ risk of higher abdominal aortic calcification scores. Subgroup analysis showed stronger association between MDS and AAC score with lower levels of magnesium
			MDS 1				
			MDS 2				
			MDS 3				
			MDS ≥4				
Luo et al. (2024) [32]	Adults (≥20 y) enrolled in the 2005–2014 NHANES data cycles (*n* = 20,585), 5 cycles, mean	48.8	MDS 0	NR	Sleep quality (trouble, disorder, duration)	NA	↑ MDS associated with ↑ sleep trouble, ↑ sleep disorder, ↑ particularly sleep apnea, and ↑ sleep duration, with no effects on incidence of insomnia and restless leg. No association between MDS grade and insufficient sleep
			MDS 1				
			MDS 2				
			MDS 3				
			MDS ≥4				
Ma (2025) [33]	Participants enrolled in the 2015–2018 NHANES data cycles (*n* = 5901); Mendelian randomization analysis		MDS ≤1	NR	OSA	3127	Noted a significant positive relationship between MDS and the risk of OSA, suggesting a causal relationship between magnesium and OSA
			MDS >1				
Ma et al. (2025) [34]	Adults (≥20 y) enrolled in the 2007–2016 NHANES data cycles (*n* = 8285)	56.8 ± 10.7	MDS 0–1	NR	Klotho levels (klotho levels as a significant marker of the aging process)	NA	Negative association between MDS and klotho levels
			MDS 2				
			MDS >2				
Ma et al. (2025) [35]	Adults (≥40y) enrolled in the 2005–2018 NHANES data cycles (*n* = 19,394) for OA incidence; with OA follow-up information (*n* = 3250) for OA mortality	57.22 ± 0.16	Continuous MDS and MDS 0	305.46 ± 1.92	Incidence of OA	3256	MDS is positively correlated with the incidence of OA
			MDS 1				
			MDS 2				
			MDS >3				
Peng et al. (2024) [36]	Adults (≥20 y) enrolled in the 2017–2020 NHANES data cycles (*n* = 3377)		MDS 0	312.67 ± 158.81	MASLD	1793	↑ MDS associated with ↑ risk of MASLD
			MDS 1				
			MDS ≥2				
Tan et al. (2024) [37]	Adults (≥20 y) enrolled in the 2007–2018 NHANES data cycles (*n* = 9708)		MDS 0–1	NR	Hypertension	4220	↑ MDS associated with ↑ risk of hypertension
			MDS 2				MDS for non-hypertensives: 0.53 ± 0.02
			MDS ≥3				MDS for hypertensives: 1.29 ± 0.03
Tian et al. (2024) [38]	Adults (≥20 y) enrolled in the 2011–2018 NHANES data cycles (*n* = 18,853)		MDS <2	NR	Diabetes	3710	MDS ≥2 associated with ↑ risk of diabetes
			MDS ≥2				Per-SD ↑ in magnesium intake was associated with ↓ risk of diabetes in adults with a MDS <2 and ≥2
Wang et al. (2022) [9]	Adults (≥20 y) enrolled in the 2005–2018 NHANES data cycles (*n* = 14,566), 5 cycles		MDS 0	304.5 ± 126.5	Osteoporosis	998	↑ MDS associated with ↑ risk of osteoporosis, particularly among individuals with suboptimal dietary magnesium intake
			MDS 1				
			MDS 2				
			MDS ≥3				
Wang et al. (2024) [39]	Adults (age NR) enrolled in the 2003–2018 NHANES data cycles (*n* = 15,565)	43.3 ± 0.3	MDS 0	NR	Metabolic syndrome	5438	↑ MDS associated with ↑ risk of metabolic syndrome
			MDS 1				
			MDS 2				
			MDS 3				
			MDS 4				
			MDS 5				
Wang et al. (2024) [40]	Adults (≥20 y) with a PIR ≤1.3 enrolled in the 2007–2018 NHANES data cycles (*n* = 7600), 6 cycles		MDS 0	269.86 ± 128.14	Kidney stones	726 (calculated from %)	↑ MDS associated with ↑ risk of kidney stones in individuals with low PIR (≤1.3)
			MDS 1				
			MDS 2				
			MDS ≥3				
Wang et al. (2024) [41]	Adults (age NR) enrolled in the 2001–2018 NHANES data cycles (*n* = 39,852)		MDS 0	NR	COPD	1762	↑ MDS associated with ↑ incidence of COPD. Dietary magnesium did not impact association
			MDS 1				
			MDS 2				
			MDS ≥3				
Wu et al. (2024) [42]	Adults (≥30 y) enrolled in the 2009–2014 NHANES data cycles (*n* = 8628)		MDS 0	315.5 ± 2.47	Periodontitis	NR	↑ MDS associated with ↑ risk of moderate/severe periodontitis and ↑ stage III/IV periodontitis
			MDS 1				
			MDS 2				
			MDS >2				
Xia (2025) [43]	Participants who complete the Questionnaire on Kidney Conditions enrolled in the 2005–2018 NHANES data cycles (*n* = 16,197)	48.57 ± 0.26	Continuous MDS and MDS 0	267.46 ± 1.82	UI	6881	Significant positive association between MDS and the prevalence of UI
			MDS 1				
			MDS 2				
			MDS >3				
Xiao et al. (2025) [44]	Adults (≥40 y) enrolled in the 2007–2018 NHANES data cycles (*n* = 18,761)		MDS 0	NR	Hyperuricemia	3484	↑ MDS was significantly associated with an increased prevalence of hyperuricemia
			MDS 1				
			MDS 2				
			MDS 3				
			MDS 4				
			MDS 5				
Xu et al. (2024) [45]	Women (≥18 y) enrolled in the 2007–2020 NHANES data cycles (*n* = 19,654)	53.48	MDS 0	301.94 (295.19–308.68)	Kidney stone disease	NR	↑ MDS associated with ↑ risk of kidney stone disease, particularly in females
			MDS 1				
			MDS 2				
			MDS 3				
			MDS 4				
			MDS 5				
Xu et al. (2024) [46]	Adults (≥40 y) enrolled in the 2005–2008 NHANES data cycles (*n* = 4953), 2 cycles	56.37	MDS ≤2	320.44 ± 6.81	Retinopathy	602	Serum 25(OH)D ≤30 nmol/L and MDS >2 associated with ↑ risk of retinopathy. Protective effect of vitamin D was primarily in those with inadequate magnesium intakes
			MDS >2				
			s-25(OH)D ≤30 nmol/L				
			s-25(OH)D >30 nmol/L				
Ye et al. (2023) [47]	Adults (≥20 y) enrolled in the 1999–2018 NHANES data cycles (*n* = 42,711), 10 cycles	47.61 (0.91)	MDS 0	299.43 ± 1.56	CVD	5015	↑ MDS associated with ↑ risk of self-reported CVD
			MDS 1				
			MDS 2				
			MDS ≥3				
Yuan et al. (2025) [48]	Adults (>18 y) enrolled in the 1999–2018 NHANES data cycles (*n* = 44,588), 10 cycles	46.88	MDS 0–1	293.14	Stroke	1751	↑ MDS was significantly associated with ↑ stroke risk in a dose-dependent manner
			MDS 2				
			MDS 3–5				
Zhao and Jin (2024) [49]	Participants (≥20 y) enrolled in the 2009–2018 NHANES data cycles (*n* = 13,197)		Low: 0 points	304.46 ± 138.05	Depression	1114	↑ MDS associated with ↑ risk of depression. Dietary magnesium had no sign impact on association on subgroup analysis
			Medium: 1–2 points				
			High: ≥3 points				
Zhao et al. (2024) [50]	Adults enrolled in the 2007–2016 NHANES data cycles (*n* = 19,227)	48.06 ± 0.27	MDS 0–1	305.14 ± 2.15	CHF	557	↑ MDS associated with ↑ risk of CHF
			MDS 2				
			MDS ≥3				
Zhao et al. (2024) [51]	Adults (≥20 y) enrolled in the 2005–2018 NHANES data cycles (*n* = 30,490)		MDS 0	NR	COPD	NR	↑ MDS associated with ↑ risk of COPD mediated by systemic inflammatory markers
			MDS 1				
			MDS 2				
			MDS 3				
			MDS 4				
			MDS 5				
Zhuang et al. (2025) [52]	Adults (40–79 y) enrolled in the 2007–2016 NHANES data cycles (*n* = 11,387)	56.25 ± 0.16	MDS continuous	306.77 ± 2.56	Serum antiaging protein klotho	NA	MDS showed a significant inverse association with serum klotho levels; compared to the low group, both middle and high MDS groups demonstrated progressively lower serum klotho levels after adjusting for all covariates
			MDS 0–1				
			MDS 2				
			MDS 3–5				
Zhou and Yao (2025) [53]	Adults (≥20 y) with DKD enrolled in the 1999–2018 NHANES data cycles (*n* = 3091)	63.98 (63.29, 64.67)	MDS 0	260.73	All-cause mortality in DKD	1373	MDS is positively associated with the prevalence of CVD in patients with DKD
			MDS 1				
			MDS 2		CVD mortality	497	
			MDS >3				

Values are means ± SD or medians (IQRs) unless specified otherwise. Upward arrows (↑) indicate increase; downward arrows (↓) indicate decrease. Abbreviations: AAC = abdominal aortic calcification; CHF = congestive heart failure; CKD = chronic kidney disease; COPD = chronic obstructive pulmonary disease; CVD = cardiovascular disease; DKD = diabetic kidney disease; hs-CRP = high-sensitivity C-reactive protein; KSR = kidney stone risk; MASLD = metabolic dysfunction-associated fatty liver disease; MDS = magnesium depletion score; NHANES = US National Health and Nutrition Examination and Survey; NR = not reported; OA = osteoarthritis; OSA = obstructive sleep apnea; PAD = peripheral artery disease; PHQ-9, 9-item Patient Health Questionnaire; PIR = poverty-to-income ratio; RA = rheumatoid arthritis; UI = urinary incontinence.

**Table 2 nutrients-17-03286-t002:** Characteristics of included prospective analyses and randomized controlled trials using the MDS.

Reference	Sample Size and Population	Age (y), Mean (SD) or Median (Range)	Intervention or Exposure (MDS)	Magnesium Intake (mg/d)	Duration and Follow-Up	Main Outcomes	No. of Events	Main Findings
Ding et al. (2025) [54]	Adults (≥18 y) with NAFLD enrolled in 1999–2018 NHANES data cycles (*n* = 16,394), 10 cycles	47.06 (0.20)	MDS 0–1	312 ± 2.09	14 y (median)	All-cause mortality	2783	↑ MDS with ↑ all-cause, cancer, and CVD mortality. Each 1-point ↑ in MDS was associated with 22% higher risk in all-cause mortality
			MDS 2			Cancer mortality	638	
			MDS 3-5			CVD mortality	1509	
Fan et al. (2021) [13]	Adults (≥20 y) enrolled in the 2005–2014 NHANES data cycles (*n* = 10,049), validation study of MDS	NR	MDS 0	NR	68.3 mo (median)	All-cause mortality	823	Low magnesium intake associated with ↑ risk of all-cause and CVD mortality among individuals with a MDS ≥2 only
			MDS 1					
			MDS 2			Cardiovascular mortality	160	
			MDS >2					
Fan et al. (2025) [55]	Adults (≥20 y) with asthma enrolled in the 2005–2018 NHANES data cycles (*n* = 4757)	45.42 ± 0.34	MDS 0	300.97 ± 4.05	Deaths until December 2019	All-cause mortality	NR	↑ MDS associated with ↑ risk of all-cause and CVD mortality
			MDS 1					
			MDS 2			CVD mortality		
			MDS ≥3					
Fan et al. (2025) [56]	Adults (age 20–74 y) with MASLD or metabolic and MetALD enrolled in the 1988–1994 NHANES III data cycles (*n* = 3802)	39.7 ± 0.4	MDS 0	NR	26 y (median)	All-cause mortality	1638	↑ MDS associated with ↑ risk of all-cause and CVD mortality. No association of MDS with risk of cancer mortality
			MDS 1			CVD mortality	542	
			MDS 2			Cancer mortality	360	
			MDS >2					
Jiang et al. [26]	Older adults (>60 y) with frailty enrolled in the 1999–2018 NHANES data cycles (*n* = 4462)	71.50 ± 0.18	MDS 0	264.03 ± 2.67	70 mo (median)	All-cause mortality	2195	↑ in CVD mortality in older adults, especially those that are inactive
			MDS 1					
			MDS 2			CVD mortality	NR	
			MDS ≥3					
Ma et al. (2025) [35]	Adults (≥40 y) enrolled in the 2005–2018 NHANES data cycles; with OA follow-up information (*n* = 3250) for OA mortality	57.22 ± 0.16	Continuous MDS and MDS 0	294.60 ± 3.46	NR	All-cause mortality	630	MDS is positively correlated with the mortality of OA. A 1-unit rise in MDS was significantly linked to an increased risk of mortality
			MDS 1					
			MDS 2			CVD mortality	172	
			MDS >3					
Song et al. (2025) [57]	Adults (≥20 y) with hypertension enrolled in the 2003–2018 NHANES data cycles (*n* = 12,485)	57.36 (0.26)	MDS 0–1	290.35 ± 2.13	90 mo median follow-up	All-cause mortality	2537	↑ MDS associated with ↑ risk of all-cause and CVD mortality among adults with hypertension
			MDS 2			CVD mortality	707	
			MDS ≥3					
Sun et al. (2024) [58]	Adults (≥20 y) with CHF enrolled in the 2007–2018 NHANES data cycles (*n* = 1022) with serum vitamin D levels	66.19 (0.29)	MDS ≤2	307.14 ± 8.86	67.53 ± 2.25 mo	All-cause mortality	418	MDS >2 associated with ↑ risk of all-cause and CVD mortality. Compared to patients with high s-25(OH)D and ≤2 MDS, those with low s-25(OH)D and MDS >2 had an ↑ risk of all-cause and CVD mortality
			MDS >2			CVD mortality	NR	
Xia (2025) [43]	Participants who complete the Questionnaire on Kidney Conditions enrolled in the 2005–2018 NHANES data cycles who had UI (*n* = 6867)		Continuous MDS and MDS 0		148 mo (median) was for survival time in MDS >3; median follow-up time was 92 mo	All-cause mortality	767	Elevated MDS levels are linked to an increased risk of all-cause mortality among patients suffering from UI
			MDS 1					
			MDS 2					
			MDS >3					
Xing et al. (2025) [59]	Adults (>18 y) with DKD enrolled in the 1988–2018 NHANES data cycles (*n* = 3179)	71 (14)	MDS 0	224 ± 133	Through December 2018	All-cause mortality	2052	↑ MDS significantly associated with all-cause mortality and CVD mortality, particularly in individuals >60 y
			MDS 1					
			MDS 2			Cause-specific mortality (CVD, malignant neoplasms, diabetes mellitus, cerebrovascular disease, and lower respiratory infection)	NR	
			MDS ≥3					
Ye et al. (2023) [47]	Adults (≥20 y) with CVD enrolled in the 1999–2018 NHANES data cycles (*n* = 5011)	64.57	MDS 0	266.88 ± 3.06	81 mo (median)	All-cause mortality	2285	↑ MDS associated with ↑ risk of all-cause and CVD mortality
			MDS 1					
			MDS 2			CVD mortality	927	
			MDS ≥3					
Yin et al. (2023) [60]	Adults (≥18 y) with CKD enrolled in the 1999–2015 NHANES data cycles (*n* = 4322)		MDS ≤2	NR	75 mo (median)	All-cause mortality	1300	MDS >2 associated with ↑ risk of all-cause and CVD mortality. No association of MDS with risk of cancer mortality. Used propensity score matching. Subgroup analyses showed MDS >2 increased all-cause and CVD mortality only in patients with inadequate magnesium intake
			MDS >2			CVD mortality	294	
						Cancer mortality	202	
Yuan et al. (2025) [48]	Adults (>18 y) with stroke enrolled in the 1999–2018 NHANES data cycles (*n* = 1751)	64.24	MDS 0–1	252.97	Deaths until December 2019	All-cause mortality	NR	↑ MDS is associated with higher all-cause mortality. Participants with high MDS had a 1.73-fold increased risk of all-cause deaths and a 2.01-fold ↑ risk of CVD deaths compared to those with none-to-low MDS
			MDS 2			CVD mortality	NR	
			MDS 3–5					
Zhang et al. (2025) [61]	Adults (≥20 y) with diabetes enrolled in the 2003–2018 NHANES data cycles (*n* = 5219)	59.26	MDS 0–1	281.88 ± 115.38	81 mo (median)	All-cause mortality	1212	↑ MDS associated with ↑ risk of all-cause and CVD mortality among adults with diabetes. The risk of all-cause mortality was higher in patients <60
			MDS 2			CVD mortality	348	
			MDS ≥3					
Zhou and Yao (2025) [53]	Adults (≥20 y) with DKD enrolled in the 1999–2018 NHANES data cycles (*n* = 3195)	64.15	MDS 0	260.73	87.2 mo (median)	All-cause mortality in DKD	1373	High MDS was associated with an elevated risk of all-cause and CVD mortality in DKD patients
			MDS 1			CVD mortality	497	
			MDS 2					
			MDS >3					
Fan et al. (2021) [13], secondary analysis of an RCT (NCT10005169)	Participants (62 ± 8.3 y) enrolled in the Personalized Prevention of Colorectal Cancer Trial who completed and had a valid magnesium tolerance test at the end of the trial (*n* = 77)	62 ± 8.3	Personalized magnesium glycinate supplementation to reduce the calcium to magnesium ratio to ~2.3	NR	12 wk	Body magnesium status		MDS (particularly when adjusted for sex and age) was validated in predicting body magnesium status

Values are means ± SD or medians (IQRs) unless specified otherwise. Upward arrows (↑) indicate increase. Abbreviations: CHF = congestive heart failure; CKD = chronic kidney disease; CVD = cardiovascular disease; DKD = diabetic kidney disease; MASLD = metabolic dysfunction-associated fatty liver disease; MDS = magnesium depletion score; MetALD = metabolic and alcohol-associated liver disease; NAFLD, nonalcoholic fatty liver disease; NHANES = US National Health and Nutrition Examination and Survey; NR = not reported; OA = osteoarthritis; RCT = randomized controlled trial; UI = urinary incontinence.

**Table 3 nutrients-17-03286-t003:** Magnesium depletion score outcome measures.

Outcome	No. of Analyses
**Cross-sectional analyses**	
Anemia	1
Arthritis or osteoporosis	3
Biomarker: high-sensitivity C-reactive protein	1
Congestive heart failure	1
Chronic obstructive pulmonary disease	2
Cardiovascular disease, hypertension, or peripheral artery disease	5
Depression	2
Frailty or aging	3
Gout, hyperuricemia, or kidney stones	6
Metabolic dysfunction or diabetes	6
Parkinson’s disease	1
Periodontitis	1
Prostate cancer	2
Sleep quality	2
Stroke	2
Urinary	2
**Prospective analyses ^a^**	
All-cause mortality	15
Cancer mortality	3
Cardiovascular mortality	13
Other cause-specific mortality (diabetes mellitus, cerebrovascular disease, lower respiratory infection)	1

**^a^** Seven analyses report both cross-sectional and longitudinal data.

**Table 4 nutrients-17-03286-t004:** Relationship of dietary magnesium intake impact on health outcomes.

Reference	Magnesium Intake (mg/d) for All Participants	Dietary Intake Methodology	Intake by Disease Subgroup (g/d)	Intake by MDS (mg/d)	Impact of Dietary Magnesium on Outcome
**Cross-sectional analyses**					
Cai et al. (2024) [18]	308.61 ± 2.03	2–24 h dietary recalls. Reported <EAR, EAR-RDA, and >RDA intakes	Nondepressed: 312.38 ± 2.04	NR	No comment
			Depressed: 265.57 ± 3.61		
Cai et al. (2025) [19]	285.10 ± 3.06	2–24 h dietary recalls. Reported <EAR, EAR-RDA, and >RDA intakes	No PAD: 287.38 ± 3.15	NR	No comment
			PAD: 241.62 ± 6.29		
Cao et al. (2024) [20]	273.0 (198.0–371.0)	1–24 h dietary recall	No gout: 273.0	NR	Dietary magnesium intake did not moderate the correlation between MDS and gout risk
			Gout: 265.0		
Cen et al. (2024) [21]	298.70 ± 2.28	2–24 h dietary recalls	Non-PD: 299.10 ± 2.29	NR	An ↑ in dietary magnesium intake was associated with a very slight ↓ in the odds of PD. Individuals in the middle and high MDS groups were at a higher risk of PD, while higher dietary magnesium intake (>250 mg) was associated with a lower risk of PD
			PD: 263.42 ± 9.68		
Chen et al. (2023) [22]	NR	2–24 h dietary recalls. Intakes reported in tertiles: Q1: ≤177.5, Q2: 177.6–316.0, and Q3: >316.1. *Included dietary supplement intake* but not defined	Non-DR: 259.1 ± 113.6	NR	↑ Dietary magnesium was linked to a ↓ incidence of DR, and magnesium supplementation was noted to be beneficial to DR prevention
			DR: 269.8 ± 113.2		
Fan et al. (2021) [13,55]	NR	2–24 h dietary recalls. ≥RDA, ≥EAR < RDA, and <EAR at 2 levels. *Included 30 d dietary supplement intake*	NR	Total magnesium intake, median (Q1–Q3)	Low magnesium intake was longitudinally associated with ↑ risks of total and CVD mortality only among those with magnesium deficiency predicted by MDS
				MDS 0: 286 (210–377)	
				MDS 1: 284 (220–374)	
				MDS 2: 283 (211–380)	
				MDS >2: 255 (198–349)	
Feng et al. (2024) [23]	275.53 ± 3.16	NR	No stroke: 278.55 ± 3.28	Low MDS: 298.48 ± 6.04	Lower dietary magnesium intake and higher MDSs were significantly associated with stroke risk
			Stroke: 243.42 ± 8.13	Medium MDS: 271.52 ± 5.10	
				High MDS: 247.23 ± 4.16	
Jiang et al. (2025) [26]	264.03 ± 2.67	NR	NR	MDS 0: 272.78 ± 9.11	No comment
				MDS 1: 260.28 ± 5.62	
				MDS 2: 263.66 ± 0.99	
				MDS 3: 265.66 ± 4.65	
				MDS ≥3: 265.66 ± 4.65	
Li et al. (2024) [27]	267.00 (203.50–352.00)	2–24 h dietary recalls. Reported as IQR	No uricemia: 270.50 (206.5–355.5)	NR	No comment
			Hyperuricemia: 255.00 (192.9–337.5)		
Li et al. (2024) [28]	NR	2–24 h dietary recalls. Sub-analysis of participants with intakes <420 and ≥420 mg/d	NR	By percent of participants with intakes >420	No comment
				MDS total: 16.5%	
				MDS 0: 19.6%	
				MDS 1: 16.7% MDS 2: 15.4%	
				MDS 3: 7.9%	
				MDS ≥4: 4.9%	
Liu et al. (2024) [29]	NR	2–24 h dietary recalls. Reported as <RDI, ≥RDI and <UL, and ≥UL	By percentage of participants: <RDI, ≥RDI and <UL, and ≥UL	By percent of participants: <RDI, ≥RDI and <UL, and ≥UL	No comment
			No anemia: 65%, 34%, and 1%	MDS 0–1: 60%, 39%, and <2.0%	
			With anemia: 71%, 28%, and <1.0%	MDS 2: 73%, 26%, and <1.0% MDS 3–5: 74%, 26%, and <1.0%	
Lu et al. (2023) [31]	NR	420 mg used as stratification for subgroup analysis	NR	MDS 0: 299.64	Proper magnesium intake may be beneficial to lower the risk of AAC in adults with a Magnesium deficiency status
				MDS 1: 314.11	
				MDS 2: 292.63	
				MDS 3: 280.83	
				MDS ≥4: 256.60	
Luo et al. (2024) [32]	NR	2–24 h dietary recalls. Stratification for subgroup analysis: <420 mg/d (85.7% of participants) and >420 mg/d (14.3% of participants)	NR	By percent of participants with intakes ≥420 mg/d	Adequate magnesium intake may be beneficial in mitigating the association of ↑ MDS and sleep disorders
				MDS 0: 18%	
				MDS 1: 17%	
				MDS 2: 15%	
				MDS 3: 9%	
				MDS ≥4 6%	
Ma et al. (2025) [34]	NR	2–24 h dietary recalls. Magnesium intake divided into tertiles (low, medium, and high) for subgroup analysis. Individual means not reported. Q1: ≤177.50, Q2: 177.6–316.0, and Q3: >3.16.1 mg/d	NR	NR	The association between dietary magnesium intake and klotho did not reach statistical significance
Ma et al. (2025) [35]	305.46 ± 1.92	Dietary recall	No OA: 308.01 ± 2.17	NR	No comment
			OA: 294.60 ± 3.46		
Peng et al. (2024) [36]	NR	1–24 h dietary recall. Analyzed data by tertiles: Q1: <230, Q2: 230–340, and Q3: >340. Includes *dietary supplement intake*s (yes or no)	NR	MDS 0: 312.67 ± 158.81 (14% DS users)	Intake was negatively associated with MAFLD only in the subgroup without magnesium deficiency
				MDS 1: 306.15 ± 148.81 (19% DS users)	
				MDS ≥2: 287.97 ± 147.11 (25% DS users)	
Tan et al. (2024) [37]	NR	2–24 h dietary recalls	No hypertension: 311.15 ± 3.65	Low MDS (0–1); 310.79 ± 3.32	No comment
			Hypertension: 294.54 ± 3.48	Medium MDS (2); 297.82 ± 5.29	
				High MDS (3–5): 267.44 ± 5.41	
Tian et al. (2024) [38]	NR	2–24 h dietary recalls	No diabetes: 306.99 ± 2.44	NR	MDS maintained a positive association with diabetes across varying levels of magnesium intake
			Diabetes: 284.16 ± 4.00		
Wang et al. (2022) [9]	304.5 ± 126.5	2–24 h dietary recalls. Divided into tertiles: <RDI: 145.5, ≥RDI: 145.5–332.5, and >UL: >332.5	No osteoporosis: 304.5 ± 126.5	NR	In subgroup analyses based on dietary magnesium intake levels, this study found that MDS positively correlated with osteoporosis in the low and middle dietary magnesium intake levels
			With osteoporosis: 263.1 ± 114.3		
Wang et al. (2024) [40]	269.86 ± 128.14 (all)	NR	NR	MDS 0: 273.08 ± 130.72	No comment
				MDS 1: 280.10 ± 133.70	
				MDS 2: 252.40 ± 115.31	
				MDS ≥3: 237.68 ± 97.61	
Wang et al. (2024) [41]	NR	2–24 h dietary recalls. Sub-analysis of participant median intake <264.5 and ≥264.5	NR	NR	Dietary magnesium did not modulate the strong correlation between MDS and COPD incidence
Wu et al. (2024) [42]	315.50 ± 2.47	NR	NR	MDS 0: 318.04 ± 3.52	No comment
				MDS 1: 322.20 ± 3.21	
				MDS 2: 309.00 ± 5.18	
				MDS >2: 276.06 ± 6.66	
Xia (2025) [43]	267.46 ± 1.82	2–24 h dietary recalls	NR	MDS 0: 263.86 ± 2.45	No comment
				MDS 1: 280.13 ± 2.69	
				MDS 2: 263.93 ± 2.91	
				MDS ≥3: 237.26 ± 3.93	
Xu et al. (2024) [45]	301.94	2–24 h dietary recalls. Intakes divided by RDA and EAR for men and women. *Included 30 d dietary supplement intake*	NR	MDS 0: 304.69	No comment
				MDS 1: 304.97	
				MDS 2: 298.94	
				MDS 3: 284.08	
				MDS 4: 263.74	
				MDS 5: 260.06	
Xu et al. (2024) [46]	320.44 ± 6.81	Not reported	No DR: 321.05 ± 7.18	NR	The protective effect of vitamin D against retinopathy was primarily present among those with inadequate magnesium levels
			DR: 314.59 ± 12.91		
Yuan et al. (2025) [48]	293.14	2–24 h dietary recalls	No stroke: 294.34	MDS 0–1: 295.61	No comment
			Stroke: 252.97	MDS 2: 292.53	
				MDS 3–5: 260.32	
Ye et al. (2023) [47]	299.43 ± 1.56	2–24 h dietary recalls	No CVD: 302.40 ± 1.58	NR	No comment
			With CVD: 268.88 ± 3.06		
Zhao and Jin (2024) [49]	304.46 ± 138.05	NR	Nondepressed: 307.40 ± 138.52	NR	No comment
			Depressed: 272.57 ± 128.69		
Zhao et al. (2023) [50]	305.14 ± 2.15	2 dietary records. Evaluated by <EAR, RDA-EAR, and ≥RDA	No CHF: 306.16 ± 2.175	Reported by percent of participants	MDS was associated with an ↑ risk of CHF among those with dietary magnesium intake below the RDA, but not intakes above the RDA
				MDS 0–1: <EAR: 54.84%	
			CHF: 258.53 ± 5.43	MDS 2: RDA-EAR: 56.63%	
				MDS ≥3: 66.52%	
Zhou and Yao (2025) [53]	260.73	24 h recall	No CVD: 267.84	MDS 0: 278.89	No comment
				MDS 1: 286.16	
			CVD: 246.78	MDS 2: 249.72	
				MDS ≥3 243.79	
Zhuang et al. (2025) [52]	306.77 ± 2.56	2 to 24 h recall	NR	Low MDS: 312.69 ± 2.65	No comment
				Middle MDS: 301.77 ± 4.61	
				High MDS: 269.79 ± 4.33	
**Prospective analyses**					
Ding et al. (2025) [54]	312 ± 2.09	2–24 h dietary recalls	NR	MDS 0–1: 316 ± 2.18	No comment
				MDS 2: 306 ± 4.11	
				MDS 3–5: 255 ± 4.51	
Fan et al. (2021) [13]	NR	2–24 h dietary recalls. Evaluated by RDA and EAR. EAR by subgroups at or above the median and below the median. *Included 30 d dietary supplement intake*	All-cause mortality (No. of cases): ≥EAR: 243, <EAR1: 215, and <EAR2: 365	MDS 0: 286	Low magnesium intake associated with ↑ risk of all-cause and cardiovascular mortality among individuals with an MDS ≥2 only. In stratified analyses by Magnesium intake, the associations remained significant only among individuals with magnesium intake less than the EAR for total morality
				MDS 1: 284	
			CV mortality (No. of cases): ≥EAR: 39, <EAR1: 215, and <EAR2: 365	MDS 2: 283	
				MDS >2: 255	
Fan et al. (2025) [55]	300.97 ± 4.05	EAR used for classification based on male/female values. Age- and sex-specific EAR was used to classify magnesium intake	NR	MDS 0: 295.30 ± 5.00	No comment
				MDS 1: 319.35 ± 6.44	
				MDS 2: 293.07 ± 6.61	
				MDS >3: 263.37 ± 7.59	
Fan et al. (2025) [56]	NR	1–24 h dietary recall. *Included 30 d dietary supplement intake*		MDS 0: 314.5 ± 6.6	The association with ↑ all-cause and CVD mortality became stronger among participants who did not meet the EAR level of magnesium intake
				MDS 1: 325.5 ± 10.1	
				MDS 2: 323.1 ± 13.3	
				MDS >2: 274.2 ± 15.8	
Jiang et al. (2025) [26]	264.03 ± 2.67	NR	NR	MDS 0: 265.02 ± 4.79	No comment
				MDS 1: 270.59 ± 2.74	
				MDS 2: 266.59 ± 2.78	
				MDS >3: 265.87 ± 3.66	
Ma et al. (2025) [35]	294.60 ± 3.46	Dietary recall survey	NR	MDS 0: 296.05 ± 8.43	No comment
				MDS 1: 313.37 ± 5.48	
				MDS 2: 287.15 ± 5.82	
				MDS ≥3: 260.51 ± 6.17	
Song et al. (2025) [57]	290.35 ± 2.13	2–24 h dietary recalls	NR	MDS 0–1: 299.98 ± 2.70	No comment
				MDS 2: 284.59 ± 0.92	
				MDS >3: 259.68 ± 3.53	
Sun et al. (2024) [58]	307.14 ± 8.86	24 h recall. *Included 30 d dietary supplement intake*	Survivors: 308.80 ± 11.88	NR	An appropriate level of serum vitamin D and magnesium intake may be beneficial to maintain cardiovascular health, thereby improving outcome
			Deaths: 304.53 ± 12.91		
Xia (2025) [43]	270.25 ± 2.52	NR	Alive: 273.28 ± 2.65	MDS 0: 263.86 ± 2.45	No comment
				MDS 1:280.13 ± 2.69	
			Dead: 239.45 ± 5.67	MDS 2: 263.93 ± 2.91	
				MDS ≥3: 237.26 ± 3.93	
Xing et al. (2025) [59]	224 (133) (median)	24 h recall	Survivors: 224 ± 134	NR	No comment
			Non-survivors: 224 ± 133		
Yuan et al. (2025) [48]	252.97	2–24 h dietary recalls	No stroke: 294.34	NR	No comment
			Stroke: 252.97		
Ye et al. (2023) [47]	268.88 ± 3.06	24 h recalls. Subgroup risk analysis by MDS <261 mg/d vs. >261 mg/d	No CVD: 302.40 ± 1.58	Subgroup risk analysis by MDS, ≤261 mg/d vs. >261 mg/d	No comment
			CVD: 268.88 ± 3.06		
Yin et al. (2023) [60]	NR	Sub-analysis of participants based on magnesium intake inadequate vs. adequate by EAR by age, in both men and women			MDS was associated with all-cause and cardiovascular-specific mortality only in those with inadequate magnesium intake
Zhang et al. (2025) [61]	281.88 ± 115.38	2–24 h dietary recalls	NR	MDS 0–1: 295.10 ± 120.08	No comment
				MDS 2: 272.72 ± 110.21	
				MDS >3: 254.53 ± 101.20	
Zhou and Yao (2025) [53] (*n* = 1072 CVD cases)	246.78	24 h recall	No CVD: 261.84	NR	No comment
			CVD: 246.78		

Values are presented as means ± SDs or medians (IQRs) unless otherwise indicated. Upward arrows (↑) indicate increase; downward arrows (↓) indicate decrease. Abbreviations: AAC = abdominal aortic calcification; CHF = congestive heart failure; COPD = chronic obstructive pulmonary disease; CVD = cardiovascular disease; DR = diabetic retinopathy; EAR = estimated average requirement; IQR = interquartile range; MAFLD = metabolic dysfunction-associated fatty liver disease; MDS = magnesium depletion score; NR, not reported; OA = osteoarthritis; PAD = peripheral artery disease; PD = Parkinson’s disease; RDA = recommended dietary allowance; RDI = recommended dietary intake; UL = tolerable upper intake limit.

## Data Availability

The original contributions presented in this study are included in the article and Appendix A. Further inquiries can be directed to the corresponding author.

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
