# Peer review of "Magnesium Depletion Score as an Indicator of Health Risk and Nutritional Status—A Scoping Review"

_nutrients, 2025, doi:10.3390/nu17203286_

Round 1

Reviewer 1 Report

Comments and Suggestions for Authors

A very interesting review paper analyzing magnesium depletion score as an indicator of health risk and nutritional status. The abstract and conclusions lack clear recommendations for MDS that the authors mentioned in the introduction. I think they are worth mentioning in the context of increased citation opportunities for this work. 
The work is worth publishing but, besides the above, some minor errors need to be eliminated.
Line 38-40
The sentence suggesting the consumption of dried beans, should definitely be changed, as they pose a risk of death if taken literally. In times when coffee cups must be labeled "Caution HOT!", you need to be very careful about the recommendations regarding the consumption of various products.
Line 36-38
Very interesting data but you need reference here. By the way, very specific data regarding Mg's involvement in enzymatic reactions were also indicated in the abstract (lines 12-13). It would be worth referring to this in the introduction, indicating the specific source.

Author Response

Response to Reviewers
Reviewer 1
Comment: A very interesting review paper analyzing magnesium depletion score as an
indicator of health risk and nutritional status.
Response: Thank you for the comment. We have addressed the reviewer’s comments in a pointby-
point response below and within the manuscript.
Comment: The abstract and conclusions lack clear recommendations for MDS that the authors
mentioned in the introduction. I think they are worth mentioning in the context of increased
citation opportunities for this work.
Response: Thank you for the comment. We have rephrased the conclusion statement in the
abstract to match the recommendations in the introduction, per the reviewer’s suggestion.
Comment: The work is worth publishing but, besides the above, some minor errors need to be
eliminated.
Response: Thank you for the comment. We have addressed the reviewer’s comments in a pointby-
point response below and within the manuscript. Our external copy-editor has also reviewed
the manuscript for editorial, grammar, and formatting errors.
Comment: Line 38-40
The sentence suggesting the consumption of dried beans, should definitely be changed, as they
pose a risk of death if taken literally. In times when coffee cups must be labeled "Caution HOT!",
you need to be very careful about the recommendations regarding the consumption of various
products.
Response: Thank you for the comment. We have removed the word “dried” in response to the
reviewer’s caution.
Comment: Line 36-38
Very interesting data but you need reference here. By the way, very specific data regarding Mg's
involvement in enzymatic reactions were also indicated in the abstract (lines 12-13). It would be
worth referring to this in the introduction, indicating the specific source.
Response: Thank you for the comment. We have rephrased the first three lines of the
introduction to be more specific and have included appropriate citations for the statement. The
abstract has also been amended to match the information within the citations.

Reviewer 2 Report

Comments and Suggestions for Authors

The manuscript is well structured, with a transparent methodology and comprehensive literature retrieval. The findings indicate that the magnesium depletion score (MDS) may serve as a valuable proxy for identifying individuals at increased risk of suboptimal status and associated comorbidities. By integrating results from both cross-sectional and prospective cohort analyses, as well as a single secondary analysis of a randomised controlled trial, the authors provide a useful overview that highlights the potential clinical relevance of the MDS. This is a valuable manuscript; however, several improvements and clarifications are required before it can be considered for publication. My suggestions and comments are provided below.

1. In the Abstract, please replace >600 with more than 600. In addition, please add one sentence with specific numbers (e.g., 48 studies; 39 cross-sectional, 15 cohorts, 1 RCT; 45/48 reported adverse associations with higher MDS).

2. In the Materials and Methods section, please add a checklist and a flow diagram with counts at each stage.

3. The discussion section is relatively brief and should be expanded to provide a more detailed synthesis of the results. In particular, please include specific examples of the health outcomes most consistently associated with elevated MDS. Incorporating comparative insights across study designs (for example, cross-sectional versus longitudinal) would also strengthen the interpretation.

4. To enhance clarity and impact, please add a conceptual figure at the end of the discussion. This figure could summarise the proposed associations between MDS, dietary magnesium intake, and health outcomes, as well as highlight key mediating or confounding factors.

5. Finally, the Conclusions section should be reformulated. Please rename it as Conclusions and Future Research Directions. In addition to summarising the main findings, please explicitly discuss research gaps and propose avenues for further investigation.

Author Response

Reviewer 2
Comment: The manuscript is well structured, with a transparent methodology and
comprehensive literature retrieval. The findings indicate that the magnesium depletion score
(MDS) may serve as a valuable proxy for identifying individuals at increased risk of suboptimal
status and associated comorbidities. By integrating results from both cross-sectional and
prospective cohort analyses, as well as a single secondary analysis of a randomized controlled
trial, the authors provide a useful overview that highlights the potential clinical relevance of the
MDS. This is a valuable manuscript; however, several improvements and clarifications are
required before it can be considered for publication. My suggestions and comments are provided
below.
Response: Thank you for the comment. We have addressed the reviewer’s comments in a pointby-
point response below and within the manuscript.
Comment: 1. In the Abstract, please replace >600 with more than 600. In addition, please add
one sentence with specific numbers (e.g., 48 studies; 39 cross-sectional, 15 cohorts, 1 RCT;
45/48 reported adverse associations with higher MDS).
Response: Thank you for the comment. We have adjusted the abstract, per the reviewer’s
suggestion.
Comment: 2. In the Materials and Methods section, please add a checklist and a flow diagram
with counts at each stage.
Response: Thank you for the comment. A flow diagram has been added to the first paragraph of
the results, as recommended by PRISMA-ScR.
Comment: 3. The discussion section is relatively brief and should be expanded to provide a
more detailed synthesis of the results. In particular, please include specific examples of the
health outcomes most consistently associated with elevated MDS. Incorporating comparative
insights across study designs (for example, cross-sectional versus longitudinal) would also
strengthen the interpretation.
Response: Thank you for the comment. This is challenging given the limitations of the data, for
example the overlapping NHANES data cycles across several group’s analyses, prospective
studies being NHANES data solely linked to the National Death Registry vs. repeat measures
data, and the former practice limiting prospective analyses to only a handful of outcomes (allcause
mortality, cancer mortality, CVD mortality are the main ones). We have added additional
text with some re-organization of the discussion section to satisfy the reviewers comment and
clarify the above point for the reader.
Comment: 4. To enhance clarity and impact, please add a conceptual figure at the end of the
discussion. This figure could summarize the proposed associations between MDS, dietary
magnesium intake, and health outcomes, as well as highlight key mediating or confounding
factors.
Response: Thank you for the comment. We feel like that the addition of a conceptual figure
adds little value to the manuscript, as the points highlighted by the reviewer are already
adequately described in the discussion section and not easily visually illustrated.
Comment: 5. Finally, the Conclusions section should be reformulated. Please rename it as
Conclusions and Future Research Directions. In addition to summarizing the main findings,
please explicitly discuss research gaps and propose avenues for further investigation.
Response: Thank you for the comment. Journal formatting requirements preclude us from
renaming the “Conclusions” section to “Conclusions and Future Research Directions.” We have
re-written the Conclusions section to better summarize the main findings and point out that
future large clinical trials are needed to confirm our findings.

Round 2

Reviewer 2 Report

Comments and Suggestions for Authors

The authors have addressed the issues raised during the first round of peer review, and I have no further comments on the revised manuscript.